# GENEVAL: An Object-Focused Framework for Evaluating Text-to-Image Alignment

**Dhruba Ghosh**[1]      **Hannaneh Hajishirzi**[1,2] *      **Ludwig Schmidt**[1,2,3] *

[1]University of Washington      [2]Allen Institute for AI      [3]LAION

* denotes equal contribution

## Abstract

Recent breakthroughs in diffusion models, multimodal pretraining, and efficient finetuning have led to an explosion of text-to-image generative models. Given human evaluation is expensive and difficult to scale, automated methods are critical for evaluating the increasingly large number of new models. However, most current automated evaluation metrics like FID or CLIPScore only offer a holistic measure of image quality or image-text alignment, and are unsuited for fine-grained or instance-level analysis. In this paper, we introduce GENEVAL, an object-focused framework to evaluate compositional image properties such as object co-occurrence, position, count, and color. We show that current object detection models can be leveraged to evaluate text-to-image models on a variety of generation tasks with strong human agreement, and that other discriminative vision models can be linked to this pipeline to further verify properties like object color. We then evaluate several open-source text-to-image models and analyze their relative generative capabilities on our benchmark. We find that recent models demonstrate significant improvement on these tasks, though they are still lacking in complex capabilities such as spatial relations and attribute binding. Finally, we demonstrate how GENEVAL might be used to help discover existing failure modes, in order to inform development of the next generation of text-to-image models. Our code to run the GENEVAL framework is publicly available at `https://github.com/djghosh13/geneval`.

## 1   Introduction

Text-to-image (T2I) models have exploded in popularity in recent years. After the introduction of DALL-E [33], breakthroughs in diffusion models quickly led to the development of more capable T2I models like DALL-E 2 [1] and Stable Diffusion [36]. Since then, T2I models have found varied use cases in creative matters like art and research applications like training data generation. Research into parameter-efficient finetuning methods has also led to a large number of new models and finetuned checkpoints of popular models. Currently, the Huggingface Hub[1] hosts over 4,000 T2I-related models and repositories.

The current gold standard for evaluating T2I models is typically human preference comparison between models, which is costly to scale up. Thus, the increase in number of T2I models makes manual evaluation inadequate. This raises the need for *reliable*, *automated* evaluation methods. However, traditional automated evaluation methods cannot analyze compositional capabilities and lack fine-grained reporting. Metrics such as the Frechet Inception Distance [16] solely evaluate image quality without taking the prompt into account. Other common metrics such as CLIPScore [15] rely

---

[1]`https://huggingface.co/models`

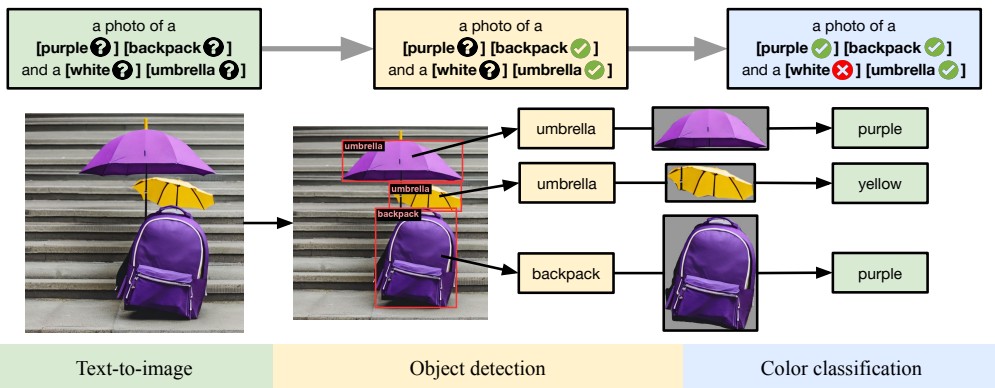

Figure 1: Visualization of GENEVAL. Modern object detection models can be used to automatically verify text-to-image generations. The detected bounding boxes and segmentation masks can be used to verify object presence, count, and position, and then passed to downstream discriminative vision models to verify fine-grained object properties such as color. The example image was generated by Stable Diffusion v2.1.

on alignment of image and text embeddings, which are not strongly correlated with human judgment on complex tasks, as our experiments show.

In light of this, we propose GENEVAL, an automated object-focused framework for evaluating T2I model capabilities on structured tasks (Figure 1). GENEVAL centers around the use of an object detection model, which verifies that the generated image contains the objects specified in the text prompt. The bounding box information and segmentation masks returned by the model are used to verify properties specified in the prompt, such as *counting* and *relative positioning* between objects. This metadata is then also passed to other vision models to evaluate other properties, in our case, *object color* classification. Overall, this results in an interpretable and modular framework which provides fine-grained information about T2I model capabilities.

We verify that our evaluation framework aligns with human judgment through a human evaluation study of 6,000 fine-grained annotations over 1,200 images. Overall, GENEVAL attains 83% agreement with annotators about the correctness of generated images, compared to the interannotator agreement of 88%. This agreement rate is boosted to 91% on images that annotators unanimously agree on, showing that our evaluation framework is highly reliable and aligns with human judgment. We also find that GENEVAL obtains significantly greater human agreement than the CLIPScore image-text alignment metric on complex tasks that involve more compositional reasoning.

We then evaluate several popular open-source T2I image models using our GENEVAL framework. Our experiments show that the new IF model [11], with a larger text encoder and pixel-space diffusion, outperforms prior Stable Diffusion models [36], with IF-XL correctly generating 61% of images over 50% from Stable Diffusion v2.1. The most recent Stable Diffusion XL [30], with various architecture and training changes, beats v2.1 on certain tasks like depicting multiple objects but fails to improve at counting. Increasing model size leads to better performance on certain tasks, but increased pretraining time does not necessarily improve performance.

Moreover, all the tested models perform poorly on complex tasks such as relative positioning and attribute binding—with the best model generating only 15% and 35% of images correctly, respectively—showing there is still much progress to be made in text-guided image generation. This supports prior claims that spatial reasoning and attribute binding are difficult for T2I models [9, 13, 14]. However, we demonstrate how the fine-grained nature of GENEVAL can inform development of new models by uncovering failure modes and patterns in current image generation.

In summary, our contributions are as follows: (1) We introduce GENEVAL, an automated framework for evaluating T2I models using existing discriminative vision models; (2) we show that current object detection models are strongly aligned with human judgment and can be used to evaluate a variety of compositional capabilities; and (3) we evaluate several popular open-source T2I models

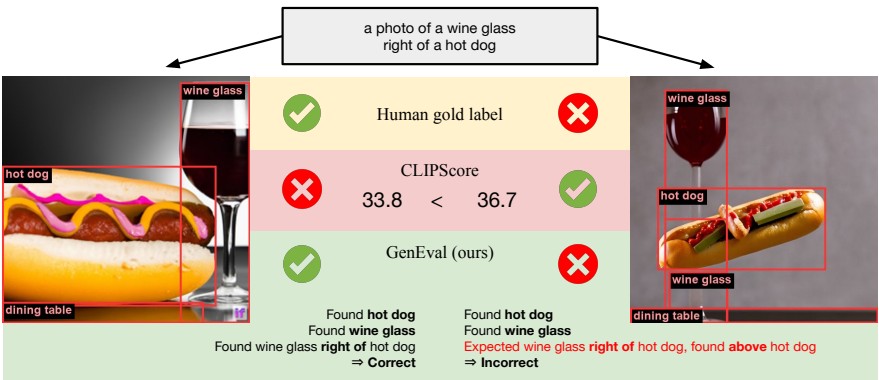

Figure 2: Comparison between GENEVAL and CLIPScore. CLIPScore returns a scalar value indicating image-text alignment, whereas GENEVAL breaks the prompt down into correct and incorrect elements before producing a final binary score. Compared to CLIPScore, GENEVAL obtains higher agreement with human judgment on complex compositional tasks.

and find significant performance improvement in recent models, though there is still much room to improve on complex compositional tasks.

## 2 Related work

**Text-to-image models.**    While T2I models have been around since 2016 [25, 35], the field saw a surge in popularity starting with the autoregressive DALL-E in 2021 [33], followed by diffusion models like DALL-E 2 [34], Midjourney [2], GLIDE [27], and Stable Diffusion [36]. Subsequent improvements scaled the text encoder and made other architectural changes, such as in Imagen [37], IF [11], and Stable Diffusion XL [30]. Meanwhile, new GAN and autoregressive models are still being developed [20, 45]. However, most popularly used models and finetuned checkpoints for T2I are still based off of the Stable Diffusion architecture.

**Automated evaluation.**    Automated T2I evaluation techniques primarily measure either image quality or image-text alignment. The Inception Score (IS) [38] and Frechet Inception Distance (FID) [16] metrics measure image quality independent of text. FID score estimates the distance between the distribution of generated images and a reference distribution of real-world images, and serves as a heuristic for how "realistic" the generated images are.

Other metrics evaluate image-text alignment. Many of these are reference-based: BLEU [28] and CIDEr [40] can be used to evaluate generations by performing captioning and comparing to a set of reference captions [17, 18], while R-Precision [29, 43] measures text recall scores from a reference dataset of captions. CLIPScore [15], on the other hand, is reference-free, and produces an alignment measure based on the cosine similarity of the prompt and generated image CLIP [31] embeddings. This is shown to be more strongly correlated with human judgment than previous reference-based metrics that compare generated captions with a set of potential captions [15].

**Human preference-based evaluation.**    Inspired by past work in NLP, several new methods evaluate generated images using models trained on direct human preference data [21, 41, 42]. This involves manual annotation of a large dataset of images — ranking groups of two or more images generated from the same prompt. A model is then trained on this dataset to predict a scalar score which correlates with human preference, offering a holistic evaluation of generated images. This serves to directly measure the endgoal of human preference, but combines multiple aspects of image generation, e.g., aesthetics, realism, and image-text alignment, into one number, whereas we aim to provide a breakdown of exact errors in each generated image.

**Object detection-based evaluation.**    A couple prior works also propose the use of object detection to evaluate T2I generation capabilities. Dall-Eval [9] trains a task-specific object detector for measuring each of several compositional reasoning tasks. They show that finetuning T2I models

on images generated from a 3D simulator can increase model performance on these tasks to some degree. In contrast, we find that with modern object detection models, training task-specific detectors is not necessary to obtain strong human agreement. This allows us to improve GenEval as better state-of-the-art vision models become available without need for further finetuning or reliance on training sets of synthetic 3D-rendered images.

Meanwhile, VISOR [14] is a different metric which allows a thorough evaluation of relative position (spatial reasoning) capabilities of T2I models. They perform a comprehensive evaluation over all triplets of object pairs and spatial relations to provide in-depth analysis of models' spatial reasoning capabilities. In contrast, we aim to cover a greater diversity of tasks, and show that object detector outputs can also be passed to downstream models that predict individual object properties such as color, expanding the scope of evaluation.

**VQA-based evaluation.**  A concurrent work, TIFA [19], demonstrates the usefulness of large language models (LLM) combined with visual question answering (VQA) models to perform fine-grained T2I evaluation. They use an LLM to generate atomic verification questions from the text prompts, and apply a VQA model to answer the question given the generated image. This is a flexible approach which can cover a diversity of prompt types, depending on the training distribution of the LLM and VQA models. Similar recent works use LLMs to evaluate generated images, either by passing in a visual description [24] or using a multimodal vision-LLM (VLLM) to directly answer questions about the image [6, 44]. In comparison, we find that GenEval outputs and failure modes are more easily interpretable, as the object detector and discriminative models produce detailed bounding box and confidence scores on a per-object basis. Furthermore, each component of our framework can be independently upgraded as better models are developed.

# 3   GenEval: Our object-focused evaluation framework

## 3.1   Setup

In order to produce a fine-grained verification of how well a generated image matches the description provided in the text prompt, we break the prompt down into the *types of objects*, their *properties* (such as color and number), and their *relations* to other objects (such as relative position).

**Text-to-image tasks.**  We focus on 6 different tasks of varying difficulty requiring various compositional skills, enumerated along with their prompts in Table 1. Here, we briefly summarize each task.

- `single object`: rendering the specified type of object. This is the simplest task, and is trivial for the modern T2I models we test.
- `two object`: rendering two different objects. This can be challenging in and of itself, as our benchmark results show (Table 2), and also serves as a base for more complex tasks like `position` and `attribute binding`.
- `counting`: rendering a specific number of one type of object. T2I models may struggle with this task, even for a small number of objects.
- `colors`: rendering an object with a specific color. We show that this can be verified reliably with a zero-shot image classifier by masking out the background.
- `position`: rendering two objects with specified positions relative to each other. Spatial understanding has been found by prior works to be challenging for T2I models [14, 9]; our findings show this is still true.
- `attribute binding`: rendering two different objects with two different colors. Prior works qualitatively observe that binding attributes like color to their respective objects is difficult for T2I models, often resulting in *swapping* (when the colors of two objects are swapped) or *leakage* (when the color instead appears on background objects) [13].

**Prompt generation.**  All our text prompts are generated from task-specific templates filled in with randomly sampled object names, colors, numbers, and relative positions. Object names are drawn from the 80 MS COCO [23] class names, with some classes renamed to remove ambiguity, e.g.,

| Task | Description | # Prompts | Template |
|------|-------------|-----------|----------|
| Single object | One object | 80 | "a photo of a/an [OBJECT]" |
| Two object | Two different objects | 99 | "a photo of a/an [OBJECT A] and a/an [OBJECT B]" |
| Counting | Specified number of an object | 80 | "a photo of [NUMBER] [OBJECT]s" |
| Colors | One object with a specified color | 94 | "a photo of a/an [COLOR] [OBJECT]" |
| Position | Two objects with specified relative position | 100 | "a photo of a/an [OBJECT A] [RELATIVE POSITION] a/an [OBJECT B]" |
| Attribute binding | Two objects with different specified colors | 100 | "a photo of a/an [COLOR A] [OBJECT A] and a/an [COLOR B] [OBJECT B]" |

Table 1: List of GENEVAL tasks. "A/an" in the templates are decided based on whether the following words starts with a vowel.

"mouse" to "computer mouse". This choice is driven by the fact that most state-of-the-art object detection models are trained on the MS COCO set of objects. Colors are taken from a list of 11 basic color terms from Berlin-Kay basic color theory [4]. For the `counting` task, the number is chosen to be either 2, 3, or 4. For the `position` task, the relative position is one of "above", "below", "to the left of", or "to the right of".

## 3.2 Evaluation framework

**Object detection.** For object detection and instance segmentation, we choose the best instance segmentation model available from the MMDetection toolbox [5], a Mask2Former [7] trained on MS COCO. For each image, we extract all detected objects above the default confidence threshold of 0.3. We find that when multiple objects of the same type are in an image (specifically, for the `counting` task), the detector tends to detect an excessive number of low confidence bounding boxes. To alleviate this, we raise the minimum confidence threshold to 0.9 for the `counting` task only. We also confirm that further pre-processing such as non-maximal suppression does not improve object detector performance in these cases. Specific details on hyperparameter decisions are provided in Appendix C.

For all tasks, we generate an intermediate score for *object presence*, marking whether the desired object types are present in the image. For the `single object` and `two object task`, this is also the final score, since the prompt only asks for the specified objects to exist. For the `counting` task, we further verify that the number of detected objects matches the number specified in the prompt. For the `position` task, we use the 2D coordinates of the detected bounding box centroids to compute relative position. We find that T2I models often generate overlapping objects; thus, we set a minimum distance along each axis (proportional to the bounding box dimensions) after which the objects will be classified as "left", "right", "above", or "below" one another.

**Color classification.** For the `colors` and `attribute binding` tasks, we use the CLIP ViT-L/14 model [31] to classify object color. For each object, using the information from the object detector, the image is cropped to the bounding box of the object. The segmentation mask is used to replace the background pixels with a solid gray background. This cropped and processed image is then passed to the CLIP model, which performs zero-shot classification between prompts of the form "a photo of a [COLOR] [OBJECT]" with all of the different candidate colors. We find that image cropping and background masking greatly improve the performance of the color classification model.

**Scoring.** For each image, the **GENEVAL score** is a binary classification of image *correctness*: whether all elements specified in the prompt were correctly rendered in the image. This score is averaged across all images generated for each task to obtain task-specific scores, and then averaged across the six tasks to produce an overall score for a given T2I model. If an image is incorrect, the framework also produces a description of how the image deviates from the expectation: whether required objects are missing, or how the computed count, position, or color of objects differs from the

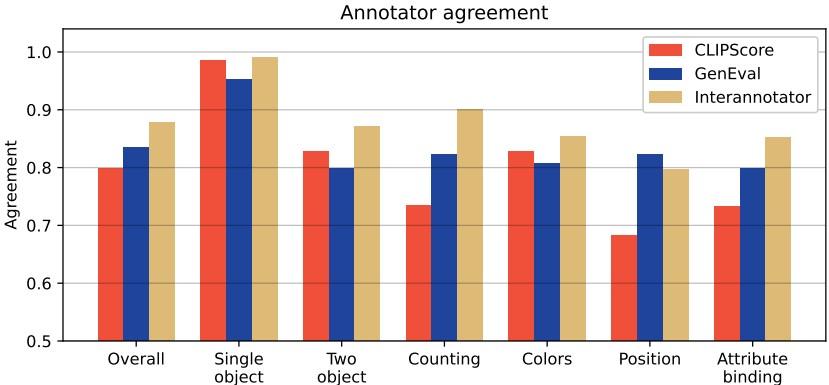

Figure 3: Human study agreement results. GENEVAL obtains higher agreement with human annotators on the more complex tasks (`counting`, `position`, and `attribute binding`) than thresholded CLIPScore, even when the CLIPScore threshold is tuned separately for each task. The difference is especially significant for the `position` task.

prompt. This can be helpful in understanding failures of both the generative T2I models (Figure 6) and the discriminative models used for evaluation (Figure 4).

## 4   Measuring alignment with human judgment

We perform a human study to verify that the object detection-based GENEVAL aligns with human perception on machine-generated images. While the Mask2Former model attains high box AP and mask AP scores on the MS COCO validation set of real-world images [5], this needs to be confirmed for AI-generated images on our distribution of prompts. We compare against interannotator agreement as well as the **CLIPScore** [15] evaluation metric, as it is also a reference-free evaluation method for judging image-text alignment. Since GENEVAL returns a binary correctness classification and CLIPScore returns a scalar cosine similarity score, we compare against human annotations by thresholding CLIPScore. For a fair comparison, we choose the best CLIPScore threshold for each task separately.

The original CLIPScore metric from [15] uses the CLIP ViT-B/32 model from [31] to compute image and text embeddings. Since then, many improved CLIP models have been released from various sources. We test several of these models and opt to compare GENEVAL against CLIPScore with OpenCLIP ViT-H/14 [8], which shows the highest human agreement on our annotated samples. The comparison of our tested CLIP models is enumerated in Table 5.

**Format.**   We conduct the study through Amazon Mechanical Turk, and gather 6,000 annotations on a total of 1,200 images, with 400 images collected from each of Stable Diffusion v2.1, IF-XL, and LAION-5B with CLIP retrieval. All models and the CLIP baseline are listed in Section 5.1. For each image, annotators are asked to list all objects they can recognize. This helps ground following responses, especially since AI-generated images may be difficult to parse. Then, for each type of object in the prompt for that image, they are asked to mark how many of that object, what primary color(s), and how realistic the objects are in the image. If there are two objects in the prompt, they are also asked about the relative position between the two objects, both horizontally and vertically. Finally, they give an overall score (on a scale of 1–4) for how well the image matches the text prompt. A screenshot of the annotation interface and further details are provided in Appendix D.

**Analysis.**   The results of our human study are presented in Figure 3. Overall, we find that the object detector and color classifier strongly correlate with human perception of the images. Across all images, GENEVAL obtains 83% agreement with human annotators, where pairwise interannotator agreement is 88%. By comparison, CLIPScore obtains 80% overall agreement with human annotators. While a threshold-tuned CLIPScore has slightly higher agreement on the simpler `single object` and `colors` tasks, GENEVAL shows higher agreement on the other four, more complex tasks. This

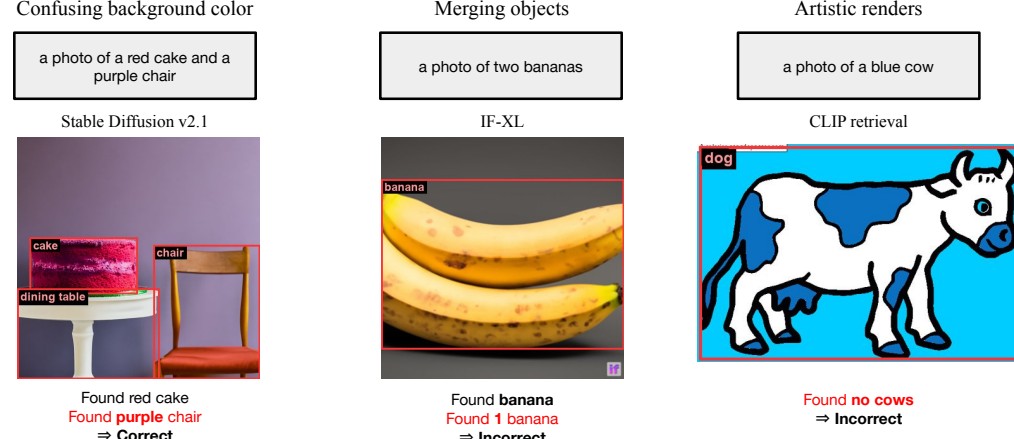

Figure 4: Failure modes of GENEVAL. (**Left**) Holes in the object which are incorporated into the segmentation mask can mislead downstream color classification. (**Center**) Images with overlapping objects of the same type are difficult for object detectors. (**Right**) Simpler artistic renderings are out-of-distribution for the detector, which reduces classification accuracy.

difference is especially pronounced in the `counting` task, where GENEVAL shows a 22 point improvement in human agreement over CLIPScore.

We also perform a qualitative analysis of the successes and failure modes of GENEVAL. An example comparison between GENEVAL and CLIPScore is shown in Figure 2. When comparing images generated by two different models from the same prompt, CLIPScore may assign significantly different scores to the images with no simple way to explain the scores. In contrast, GENEVAL outputs a sequence of verifications that explain why an image was marked correct or incorrect.

This also makes it easier to debug cases where GENEVAL does not match human judgment (Figure 4). The color classifier can be confused by objects with holes, as the segmentation masks generated by our object detector may erroneously include these holes as part of the object. Other hard cases for the object detector include multiple overlapping objects of the same type, which may result in merged bounding boxes, and out-of-distribution images such as clip art. Despite these occasional failure modes, we find that GENEVAL aligns with human judgment significantly more than the CLIPScore baseline: on the 860 examples where human scores (across 5 annotators) are unanimous, GENEVAL obtains 91% overall agreement, while CLIPScore obtains 87% overall agreement.

## 5 Benchmarking progress in recent T2I models

### 5.1 Experiments

**Models.** We evaluate a variety of open-source text-to-image models. This includes all versions of Stable Diffusion (SD) v1 and v2 [36], the recently released SD XL model, the IF pixel-space diffusion models from DeepFloyd [11], and the older model minDALL-E [22, 10], inspired by the original DALL-E model. In our primary results, we display only the latest or largest variant of each model, namely, IF-XL, SDv2.1, SDv1.5, and SD-XL 1.0. We also compare these models against a baseline of real images from LAION-5B [39], selected using CLIP ViT-L/14 image retrieval [3]. Further details are provided in Appendix C.

**Image generation.** We evaluate the models on a set of 553 prompts spanning all six tasks enumerated in Table 1. For each prompt, we generate 4 images, and the GENEVAL score is averaged over all generated images. This choice is motivated by current text-to-image APIs like DALL-E 2 [1] and Midjourney [2], which generate 4 images for a prompt to allow the user more choice. Other parameters, such as image resolution, sampling steps, and sampling method, are left at their default values for each model.

| Model | Single object | Two object | Counting | Colors | Position | Attribute binding | Overall | CLIPScore | Human |
|---|---|---|---|---|---|---|---|---|---|
| CLIP retrieval | 0.89 | 0.22 | 0.37 | 0.62 | 0.03 | 0.00 | 0.35 | 27.8 | 0.42 |
| minDALL-E | 0.73 | 0.11 | 0.12 | 0.37 | 0.02 | 0.01 | 0.23 | 27.3 | — |
| SDv1.5 | **0.97** | 0.38 | 0.35 | 0.76 | 0.04 | 0.06 | 0.43 | 33.5 | — |
| SDv2.1 | **0.98** | 0.51 | 0.44 | **0.85** | 0.07 | 0.17 | 0.50 | 36.2 | 0.57 |
| SD-XL | **0.98** | **0.74** | 0.39 | **0.85** | **0.15** | 0.23 | 0.55 | **36.7** | — |
| IF-XL | **0.97** | **0.74** | **0.66** | 0.81 | **0.13** | **0.35** | **0.61** | 36.5 | **0.72** |

Table 2: **GENEVAL scores** over the main T2I models and baseline. The recent models, IF-XL and SD-XL, display a significant improvement over previous models on challenging tasks. However, tasks like `counting`, `position`, and `attribute binding` still show much room for improvement. Overall scores vary by 0.01–0.02 across random seeds. The relative ordering of the evaluated T2I models is consistent with CLIPScore and human annotators, except with SD-XL and IF-XL, which CLIPScore cannot distinguish between.

## 5.2 Results

**Overall rankings.** Table 2 reports the results of evaluating the T2I models and CLIP baseline on the six GENEVAL tasks. Of these models, IF-XL has the best overall performance, followed by SD-XL. While the `single object` and `colors` tasks are relatively easy for all models (as expected from the complexity of the task), there is a significant performance gap between models on the other four tasks. In particular, `two object` performance has risen in the recent IF-XL and SD-XL models, while `position` and `attribute binding` tasks remain difficult overall. Unsurprisingly, the oldest model, minDALL-E, performs worst overall. Interestingly, the CLIP retrieval baseline surpasses minDALL-E and even SDv1.5 on the `counting` task, but otherwise performs poorly. This is to be expected, as many object pairings and colorings, e.g. "a photo of a blue cow", would rarely appear in a real-world dataset like LAION-5B.

**Effects of model scale.** Figure 5 compares the GENEVAL performance across the three different scales of the DeepFloyd IF models: IF-M, IF-L, and IF-XL. All three models utilize the same T5-XXL text encoder [32], with increasingly sized text-to-image and image upscaling modules. We observe that on the `two object`, `counting`, and especially the `attribute binding` tasks, we see consistent improvement as model size increases. However, we do not see such an improvement on the `position` task. This suggests that certain T2I capabilities may not be improved by vision model scale only; different training data or a better text encoder may be required.

**Effects of increased pretraining.** SDv1 is available in five checkpoints, corresponding to continued training on increasing amounts of data from the LAION-5B dataset. We evaluate all five model checkpoints and report the results in Figure 5. Surprisingly, there is little increase in model

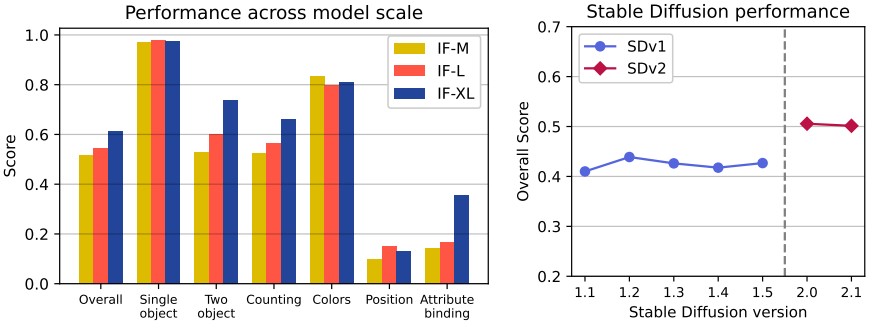

Figure 5: (**Left**) Change in model performance IF model scales. Overall, GENEVAL score increases with model size, though this does not appear to be the case for the `position` task. (**Right**) Change in model performance over successive iterations of Stable Diffusion. Despite each subsequent version having been trained for more iterations, there is no consistent performance increase from v1.1 to v1.5. In contrast, v2, which uses a different text encoder, significantly improves performance.

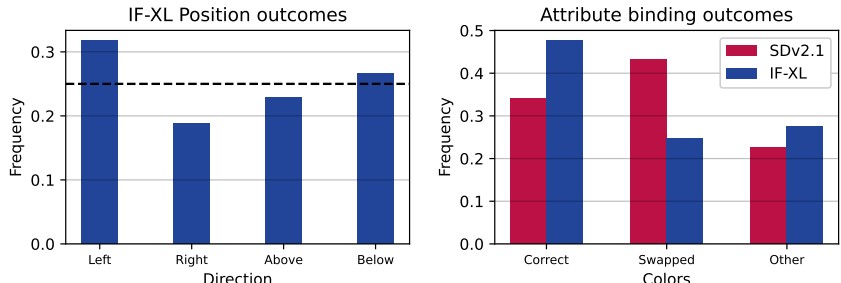

Figure 6: Failure modes of T2I models. (**Left**) IF-XL displays a position bias where the first mentioned object is more likely to be on the left than the right of the second object. (**Right**) SDv2.1 is prone to swap the colors of two objects, failing to correctly *bind* attributes to their respective objects.

performance with increased pretraining steps. While there appears to be a noticeable step up from v1.1 to v1.2, there is no further increase in overall scores after that. However, SDv2 does show significant improvement over SDv1. This may be explained by the change in text encoder: SDv2 uses OpenCLIP ViT-H/14 [8], which is larger and trained on different data compared to the CLIP ViT-L/14 [31] used by SDv1. SD-XL also increases scores over SDv2, though there are numerous qualitative differences in architecture and training that may contribute to this improvement.

**Understanding T2I model failure cases.** Our GENEVAL framework produces fine-grained output which facilitates quantitative analysis of when and how T2I models fail. We describe two interesting patterns in Figure 6. On the `position` task, it appears that when IF-XL is able to generate both objects, it is biased towards placing the first object *to the left* of the second object, and biased against placing it to the right, even when the prompts are evenly distributed among all four directions. On the `attribute binding` task, we find that Stable Diffusion v2.1 is significantly more susceptible to color *swapping* (applying one of the specified colors to the wrong object) as compared to IF-XL. These examples showcase how GENEVAL can be used to analyze flaws and biases in T2I generations, suggesting avenues for future improvement in image generation. Qualitative examples of these findings are shown in Appendix A.

# 6 Limitations

GENEVAL is primarily limited by the performance of the object detector used and the availability of discriminative vision models. We note that the best object detection algorithms are still trained or finetuned on the MS-COCO dataset, which has a limited number of classes defined at a particular level of granularity. This means that while, for example, GENEVAL can verify the number of people present in a generated image, it cannot verify the number of fingers on each person's hands. Similarly, as noted in Figure 4, object detectors trained primarily on photos do not generalize well to visually distinct art. These constraints may be removed in the future with the development of more powerful open-vocabulary object detectors trained on a wider distribution of images [26].

# 7 Conclusion

We introduce GENEVAL, a new object-focused framework for automated evaluation of text-to-image models. GENEVAL evaluates T2I model capabilities across a suite of compositional reasoning tasks using object detection and color classification to verify fine-grained object properties. We perform a human study and find that GENEVAL scores align strongly with human judgment on an instance level, beating out prior approaches measuring overall image-text alignment. We then benchmark open-source T2I models and find that while complex compositional tasks like relative position and attribute binding are difficult for current T2I models, GENEVAL can help identify failure modes to facilitate future improvement. We hope to expand GENEVAL in future work to take advantage of the wide variety of discriminative vision models that have been developed, to produce an even broader highly interpretable evaluation framework for text-to-image generation.

## Acknowledgments and Disclosure of Funding

We would like to thank Achal Dave, Vivek Ramanujan, Ellen Wu, Yushi Hu, Joongwon Kim, Taylor Sorensen, Jeffrey Li, Sebastin Santy, and many other members of UW and UW NLP for their discussion and constructive feedback. We would also like to thank Hyak computing cluster at the University of Washington for providing access to computational resources for running our experiments. This work was funded in part by the DARPA MCS program through NIWC Pacific (N66001-19-2-4031), NSF IIS-2044660, Open Philanthropy, the Allen Institute for AI, and NSF grants DMS-2134012 and CCF-2019844 as a part of NSF Institute for Foundations of Machine Learning (IFML).

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

| Model | Overall | Single object | Two object | Counting | Colors | Position | Attribute binding |
|---|---|---|---|---|---|---|---|
| CLIP retrieval | 0.35 | 0.89 | 0.22 | 0.37 | 0.62 | 0.03 | 0.00 |
| minDALL-E | 0.23 | 0.73 | 0.11 | 0.12 | 0.37 | 0.02 | 0.01 |
| SD v1.1 | 0.41 | **0.98** | 0.31 | 0.33 | 0.77 | 0.02 | 0.05 |
| SD v1.2 | 0.44 | **0.97** | 0.41 | 0.37 | 0.76 | 0.03 | 0.10 |
| SD v1.3 | 0.43 | **0.97** | 0.38 | 0.35 | 0.77 | 0.03 | 0.05 |
| SD v1.4 | 0.42 | **0.98** | 0.36 | 0.35 | 0.73 | 0.01 | 0.07 |
| SD v1.5 | 0.43 | **0.97** | 0.38 | 0.35 | 0.76 | 0.04 | 0.06 |
| SD v2.0 | 0.51 | **0.98** | 0.50 | 0.48 | **0.86** | 0.06 | 0.15 |
| SD v2.1 | 0.50 | **0.98** | 0.51 | 0.44 | **0.85** | 0.07 | 0.17 |
| SD-XL | 0.55 | **0.98** | **0.74** | 0.39 | **0.85** | **0.15** | 0.23 |
| IF-M | 0.52 | **0.97** | 0.53 | 0.53 | **0.84** | 0.10 | 0.14 |
| IF-L | 0.54 | **0.98** | 0.60 | 0.57 | 0.80 | **0.15** | 0.17 |
| IF-XL | **0.61** | **0.97** | **0.74** | **0.66** | 0.81 | **0.13** | **0.35** |

Table 3: GENEVAL scores for all models evaluated, including different versions of each model. Overall scores have a standard deviation of about 0.01 across random seeds.

# A  Further experiments

Table 3 shows per-task GENEVAL scores for all the models we evaluate. Here, we cover additional ablations and analysis.

## A.1  Alignment with human judgment

Figure 3 shows percent agreement with human judgment across tasks. The agreement rates vary across tasks partly due to differing scores across tasks, i.e., easier tasks will have higher baseline agreement. Thus, Figure 7 compares **Cohen's kappa** scores, which range from $-1$ (complete disagreement) to 1 (complete agreement) and take random agreement chances into account. `single object` kappa for GENEVAL is approximately 0, mainly because almost all images for that task are correct, which penalizes any disagreements greatly.

## A.2  Evaluation parameters

**Color classification.**     As explained in Section 3.2, our color classification involves two preprocessing steps: *cropping* the image to the bounding box, and *masking* out the background to replace it with gray. Table 4 shows the individual effects of these steps on the two color-related tasks. Both cropping

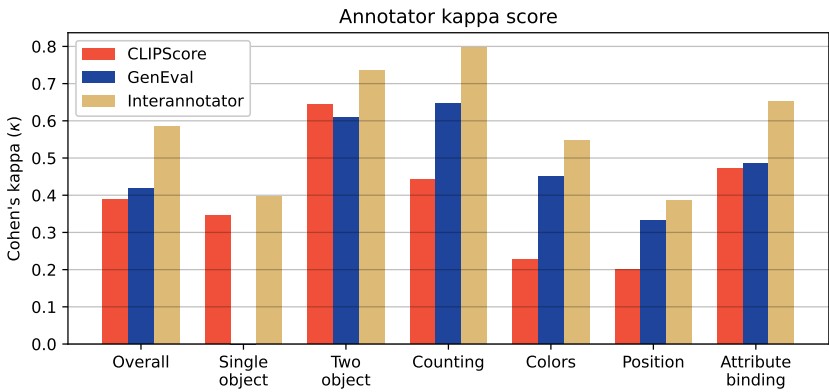

Figure 7: Human study Cohen's kappa results. CLIPscore shows strong correlation with human judgment on the single object and two object tasks, but is beaten out by GENEVAL for the other tasks. GENEVAL obtains a kappa score of 0 on the single object task because almost all annotated images were correct, penalizing disagreements more strongly.

| Method | Crop | Mask | Colors | Attribute binding |
|---|---|---|---|---|
| GENEVAL | ✗ | ✗ | 0.32 | 0.01 |
|  | ✓ | ✗ | 0.37 | 0.33 |
|  | ✗ | ✓ | 0.43 | 0.47 |
|  | ✓ | ✓ | **0.45** | **0.49** |
| CLIPScore | — | — | 0.23 | 0.47 |
| Interannotator | — | — | 0.55 | 0.65 |

Table 4: Cohen's kappa agreement with human annotators for different color classification methods. *Bounding box cropping* and *background masking* both increase alignment with human judgment, especially for the `attribute binding` task where the presence of other objects may otherwise confuse the classifier. Combining cropping with masking provides a small gain over just masking.

and masking are individually useful at removing distractions (i.e. other objects in the image), which manifests most often in the `attribute binding` task. Combining the two gives the best human agreement.

**Counting task threshold.** When images contain multiple instances of the same type of object, the object detector tends to predict disproportionately more bounding boxes. To remedy this, we increase the minimum confidence threshold for only the `counting` task (Figure 8). The optimal threshold is much higher, around 0.9, and increases Cohen's kappa agreement greatly, from 0.37 to 0.65. Using K-fold cross-validation ($k = 5$), we confirm that this is the optimal threshold across all splits, obtaining $0.823 \pm 0.013$ agreement with human annotations on validation splits.

**Position task minimum distance.** As described in Appendix C.3, there are rare cases when two objects may be generated too close together to determine their relative position. In these cases, human annotators mark the objects as neutrally/not offset from one another. We find this has an insignificant effect on overall scores, but setting a small minimum distance between objects before they are classified as to the left/right/above/below one another does improve alignment with human judgment (Figure 8). Using K-fold cross-validation ($k = 5$), we confirm that our choice of threshold is not overfit: we obtain $0.822 \pm 0.013$ agreement with human annotations on validation splits.

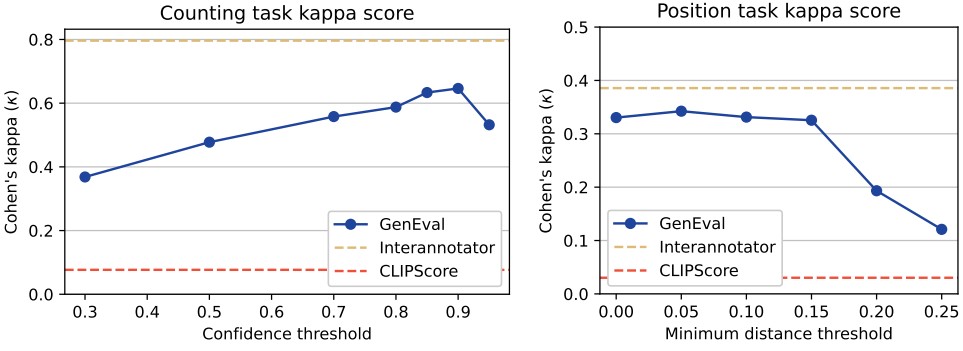

Figure 8: Cohen's kappa agreement with human annotators for varying evaluation hyperparameters. (**Left**) `counting` task confidence threshold. The default threshold of 0.3 is too low when multiple instance of the same object are in the image, resulting in superfluous detected objects. (**Right**) `position` task minimum distance threshold. In rare cases when objects are merged or very close together, they should be classified as neutral/not offset in the given direction to match human judgment.

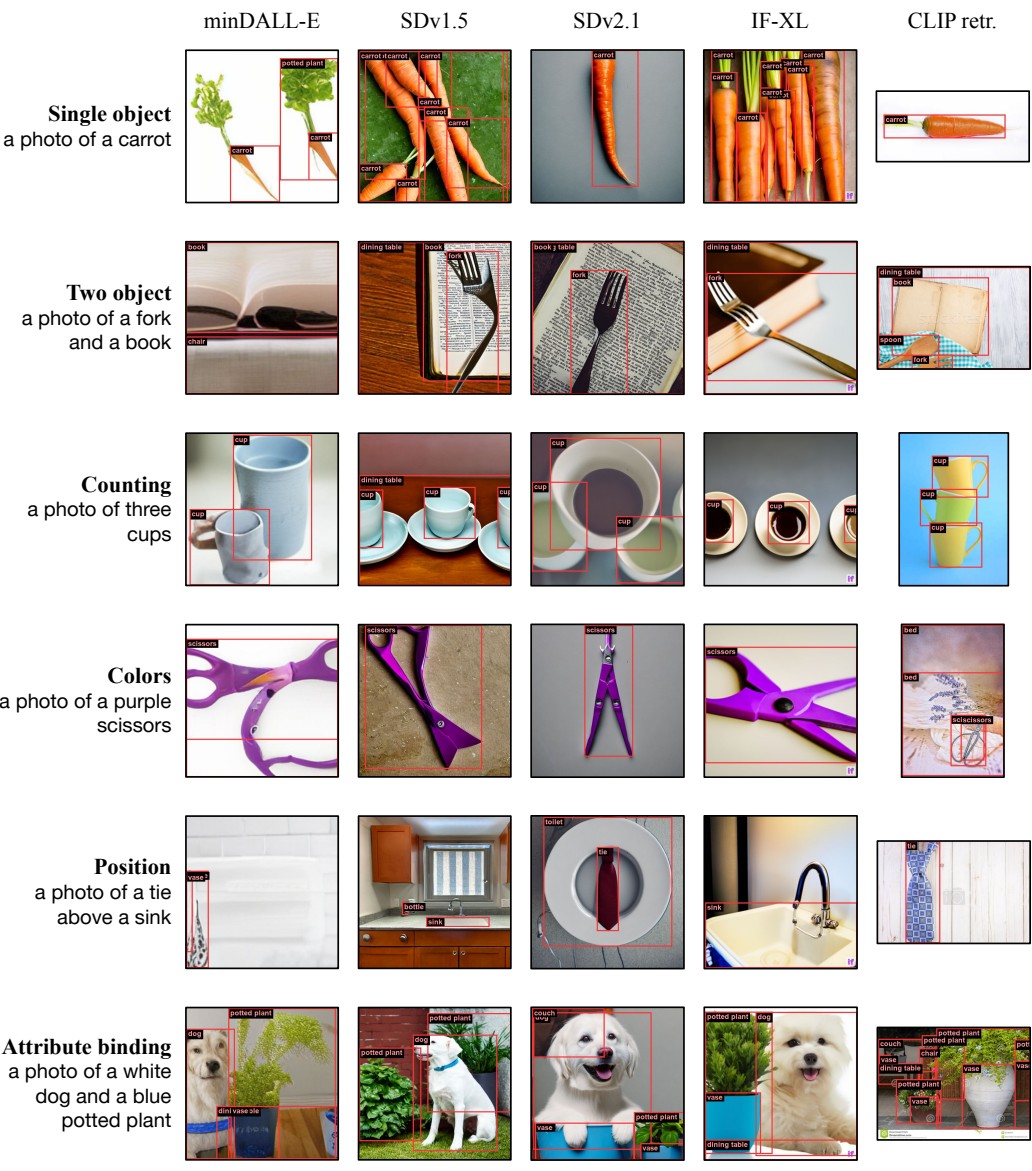

Figure 9: Random examples of images generated by each model.

## A.3 Qualitative examples

Figure 9 displays sample generated images for each model for an example prompt from each task. In general, all of the T2I models struggle with objects that have more complicated structure; items like forks (second row) and scissors (fourth row) are difficult even for Stable Diffusion and IF. Certain object combinations are also surprisingly difficult for the models to generate at all (fifth row), while others are easier, perhaps due to co-occurrence in the training data. Note, however, that the T2I models often do succeed in generating unseen or rarely seen object pairs: for example, the CLIP retrieval from LAION-5B in row 6 does not find an image of a dog and a potted plant, whereas Stable Diffusion (primarily trained on LAION) succeeds in doing so.

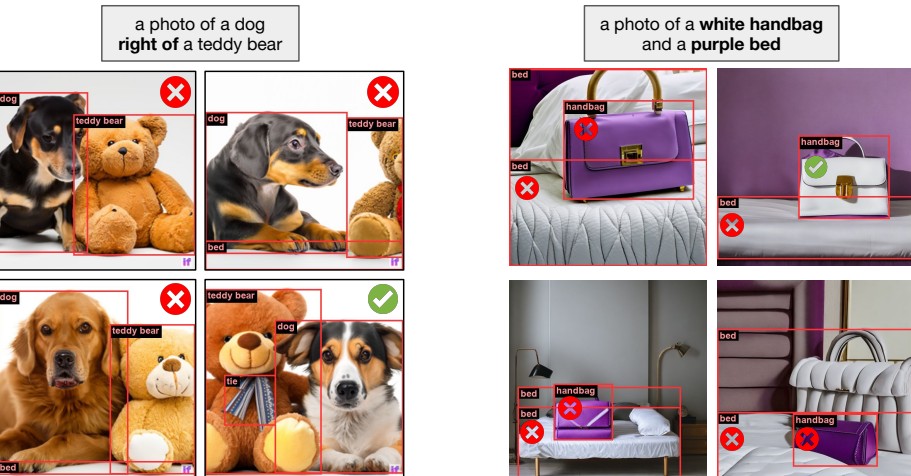

Figure 10: T2I failure modes discovered through GENEVAL. (**Left**) IF-XL exhibits a position bias where the first object is more likely to be on the *left of* the second object. (**Right**) Stable Diffusion v2.1 has a tendency to *swap* the specified colors in the `attribute binding` task, or sometimes to *leak* the specified color into the background instead, as in the top right example.

Figure 10 shows some examples of T2I failure modes as described in Section 5.2. The position bias in IF-XL and the color swapping tendency of Stable Diffusion can be discovered through quantitative analysis, and confirmed through qualitative observation of the generated images. These failure modes suggest avenues of research for future targeted improvement of T2I capabilities.

## B   Ethics statement

The primary purpose of this work is to facilitate the development of better text-to-image generative models. While we hope that such models are used for benign uses such as generating art, drafting creative ideas, or producing synthetic training data for vision models, we acknowledge that text-to-image technology can be used for harm. This may manifest through spreading misinformation with deepfakes to cause individual or widespread harm. This is a potential impact down the line for any research into developing text-to-image models, and careful consideration will be required to mitigate this harm.

Another potential issue is amplification of biases present in the evaluation method. For example, the object detector classes implicitly place importance on certain types of objects, and the detector and the color classifier encode biases present in their respective training data. However, we hope that our evaluation framework will be expanded and improved with better models, so that mitigating bias upstream, e.g., by diversifying training data and expanding the number of object classes, will also lead to a more fair evaluation.

## C   Technical details

### C.1   Prompt generation

For each task, we generate 100 random prompts and then remove duplicates, sampling uniformly (with specific exceptions) from:

- Objects: 80 MS-COCO object names
- Numbers: "two", "three", or "four"
- Positions: "above", "below", "left of", "right of"
- Colors: "red", "orange", "yellow", "green", "blue", "purple", "pink", "brown", "black", "white"

Note that "gray" is excluded from the generation colors, because we find that gray objects tend to be close to black or white, which makes those prompts ambiguous. This also enables us to fill in the background of cropped images with gray when passed to the color classifier, since we can exclude gray as a potential classification option. Furthermore, for the color-related tasks (`colors` and `attribute binding`) we exclude the "person" class from the list of candidate objects.

The prompt text is generated from the templates listed in Table 1. As stated, the choice between "a" and "an" is decided based on whether the following letter is a vowel or not. Plurals are not necessarily grammatical; an "s" is simply appended to the object name. Metadata about each prompt, required for evaluation, is stored in JSON format. For example:

```
{
  "tag": "colors",
  "include": [
    {"class": "bicycle", "count": 1, "color": "red"}
  ],
  "prompt": "a photo of a red bicycle"
}
```

## C.2  Image generation

**Text-to-image models and baseline.**    We evaluate the following models using our framework:

- **minDALL-E** [22], a port of DALL-E Mini [10], inspired by the original DALL-E [33]. minDALL-E is similarly autoregressive but replaces the discrete VAE with a VQGAN. The generated image resolution is 256x256.
- **Stable Diffusion v1** [36], with five iterations from v1.1 to v1.5. SDv1 is a latent space diffusion model using the pretrained CLIP ViT-L/14 text encoder. The default image resolution is 512x512.
- **Stable Diffusion v2** [36], with two iterations, v2.0 and v2.1. SDv2 shares a similar architecture to SDv1, but notably, uses a larger OpenCLIP ViT-H/14 text encoder trained on LAION-5B. It is also trained on larger images, so the default image resolution is 768x768.
- **IF** [11], with three sizes: IF-M, IF-L, IF-XL. IF is a three-stage pixel space diffusion model. We exclude the third stage upscaler in our evaluation because at this time, they recommend using the Stable Diffusion 4x upscaler and the IF upscaler is not available on Huggingface. There are also only two second stage sizes, M and L, so IF-XL uses the L-size stage 2 model. The output resolution of the second stage is 256x256.
- **Stable Diffusion XL** [30], which massively increases the UNet backbone size from Stable Diffusion v2 and incorporates two text encoders. There is a second refinement model, which we do not use as it does not affect the composition of the image.

As well as the following real image baseline:

- **CLIP retrieval** [3] from LAION-5B. The LAION-5B dataset is indexed with CLIP ViT-L/14, and for each prompt, we select the images with the highest image-text embedding alignment. These images are not fixed to any specific resolution or aspect ratio.

**Generation.**    For each prompt, we generate 4 images (in the case of CLIP retrieval, we select the top 4 matches). In typical usage of T2I models, there are a wide variety of parameters that might be modified to improve generations, such as sampling method, number of diffusion steps, prompt prefixes/suffixes, and negative prompts. We do not tune these hyperparameters due to the compute cost of image generation, as well as the open-ended nature of text prompts. However, some preliminary tests, such as removing the "a photo of" prefix or adding a "universal" (general purpose) negative prompt, do not seem to affect GENEVAL scores positively. We expect that careful experimentation with individual models may lead to small performance gains.

## C.3  Evaluation

**Object detection.**    We use the Mask2Former with Swin-S backbone trained for instance segmentation available from the MMDetection toolbox from OpenMMLab, released under an Apache 2.0

license. For all tasks except for `counting`, we take objects with a confidence above the default threshold of 0.3. For `counting`, we find that a higher threshold of 0.9 gives the highest human agreement results (Figure 8), particularly due to multiple bounding boxes being generated for overlapping objects of the same type. We also experiment with non-maximal suppression (NMS) within each class but find that this does not raise human agreement. For all but the `counting` task, there is also no upper limit on the number of objects. For example, for the prompt "a photo of a book and a laptop", the image is counted as correct even if there are three books and two laptops.

**Relative position.** The relative position between objects for the `position` task is evaluated from the bounding box coordinates. Prior works [9, 14] use a simple heuristic where the relative position is determined by the difference between bounding box centroids: if object A is centered at $(x_A, y_A)$ and object B is centered at $(x_B, y_B)$, then:

$$x_B > x_A \implies \text{B is right of A}$$
$$x_B < x_A \implies \text{B is left of A}$$
$$y_B > y_A \implies \text{B is below A}$$
$$y_B < y_A \implies \text{B is above A}.$$

However, qualitatively, some images generate with both objects overlapping or merged. In these cases, human annotators do not perceive the objects as being offset from one another (whereas with this heuristic, that is only the case if the objects are aligned pixel-perfectly). Thus, we add another term denoting the minimum threshold before two objects are considered "visibly offset" in a direction. If object A has dimensions $w_A \times h_A$ and object B has dimensions $w_B \times h_B$, then with a distance threshold of $c$ we have:

$$x_B > x_A + c(w_A + w_B) \implies \text{B is right of A}$$
$$x_B < x_A - c(w_A + w_B) \implies \text{B is left of A}$$
$$y_B > y_A + c(h_A + h_B) \implies \text{B is below A}$$
$$y_B > y_A - c(h_A + h_B) \implies \text{B is above A}.$$

We go with this method for evaluating relative position, and find that $c = 0.1$ optimizes human agreement.

**Color classification.** We use the CLIP ViT-L/14 model from OpenAI [31], released under an MIT license, as a zeroshot color classifier. We also test this with the ViT-B/32 model, but find it shows lower human agreement. For each candidate object, the image is cropped down to the bounding box of the object, and the segmentation mask is used to replace all background pixels with gray. We find that this shows higher agreement with human judgment than using the whole image or not replacing the background (Table 4). For the classification, we classify between the 10 candidate colors described above in C.1, with the following class templates:

- "a photo of a [COLOR] [OBJECT]"
- "a photo of a [COLOR]-colored [OBJECT]"
- "a photo of a [COLOR] object"

where [COLOR] is replaced with the color name and [OBJECT] is replaced with the detected object's name. The normalized prompt embeddings for each color are averaged, and the prompt embedding with the highest cosine similarity to the cropped image determines the predicted color.

**CLIPScore baseline.** We compare our evaluation framework against CLIPScore [15]. This boils down to the cosine similarity between the prompt and image CLIP embeddings, clipped to be at least 0. The original paper multiplies the score by 2.5 to span a wider natural range; since this is has no real impact on comparing CLIPScore values, we simply scale the similarity to be between 0 (orthogonal) and 100 (perfectly aligned).

[15] used the CLIP ViT-B/32 model from [31] available at the time; since newer improved CLIP models have been developed more recently, we test several different potential models and choose the model most aligned with human judgment as a baseline. We compare the ViT-B/32 model against the larger ViT-L/14 model [31], an even larger ViT-H/14 model from OpenCLIP [8], and the EVA-02-CLIP model from [12]. Table 5 displays the results of this comparison. Overall, the OpenCLIP

| CLIP Model | Overall | Single object | Two object | Counting | Colors | Position | Attribute binding | ImageNet-1k zeroshot |
|---|---|---|---|---|---|---|---|---|
| ViT-B/32 [31] | 0.773 | **0.991** | 0.720 | 0.604 | 0.811 | **0.790** | 0.723 | 0.632 |
| ViT-L/14 [31] | 0.760 | 0.985 | 0.692 | 0.583 | 0.796 | **0.786** | 0.719 | 0.753 |
| ViT-H/14 [8] | **0.798** | 0.986 | **0.828** | **0.734** | **0.827** | 0.682 | **0.732** | 0.780 |
| EVA-02-E/14+ [12] | 0.600 | **0.992** | 0.628 | 0.612 | 0.804 | 0.240 | 0.322 | **0.820** |
| GENEVAL | 0.834 | 0.953 | 0.799 | 0.823 | 0.808 | 0.823 | 0.799 | |
| Interannotator | 0.878 | 0.990 | 0.872 | 0.901 | 0.854 | 0.797 | 0.853 | |

Table 5: Human study agreement with CLIPScore using various CLIP models. While the OpenCLIP ViT-H/14 obtains the best overall agreement with human judgment, it is significantly worse at evaluating `position`, suggesting a qualitative difference in training data compared to [31]. In addition while EVA-02-CLIP has the highest accuracy on ImageNet, this does not appear to transfer to tasks aside from `single object` recognition.

ViT-H/14 agrees most with human annotators — however, none of the models are consistently better or worse for CLIPScore across all our tasks.

# D   Human study details

The human study was conducted through Amazon Mechanical Turk. Annotators were asked to answer a series of questions about a displayed image, as enumerated below and shown in Figure 11. There were very limited potential participant risks, if they were to be exposed to an image that was disturbing or not safe for work (NSFW). However, the images we used were sampled from templated prompts which were not in themselves offensive. Furthermore, all images generated by our models or retrieved by CLIP were passed through NSFW filters which would black out any potentially unsafe images.

We annotated 400 images generated from 100 randomly chosen prompts for each of Stable Diffusion v2.1, IF-XL, and CLIP retrieval from LAION-5B, with 5 annotations for each image. Depending on the prompt (which affected the questions asked), each annotation took about 1–2 minutes. Each crowdworker was paid an estimated $15 per hour. In total, across all 6,000 annotations and including Mechanical Turk fees, $3,600 were spent on our human study.

## D.1   Full text

The task consists of general instructions and a set of questions specific to each image. For each instance of the task, the "[OBJECT]" in each question is replaced with the names of the objects from the prompt. If the prompt consists of two types of objects (e.g., "a photo of a bird and a skateboard"), questions 2 and 3 refer to each type respectively. If the prompt only has one type of object (e.g., "a photo of four handbags"), question 3 and question 4 are skipped.

**Instructions.**   Thanks for participating in this HIT! You will view an image, potentially generated by an AI image generator. Please answer all of the following questions to the best of your abilities. For each image you may be asked about:

- List of objects: What objects are you able to discern in the image?
- Number of objects: How many objects of a particular type are visible, either fully or partially?
- Color of objects: What color(s) would you consider the object(s) to be?
- Realism: How realistic or structurally correct are the objects?
- Relative position: What is the relative position in the image of one object to another?
- Caption fit: How well does the image correspond to a provided caption?

A few notes:

- Take a look at the examples if any of the questions are unclear to you.

- Complete question 1 before moving on to other questions. If you are unable to recognize a particular object, guess what it is or use a brief description.
- Additional questions may appear based on your previous answers. Make sure to answer all questions shown.
- When asked about color, select at least one color that best matches the true color of the object(s).
- Do not consider color when deciding how realistic an object is; focus on the shape and form compared to real life.
- When asked about position, an object can be both above/below and to the left/right of another object. Mark position from your perspective looking at the 2D image, not from the camera perspective.
- The image may be just a black square, in which case, you should say there are no objects.
- Please answer with care: Some HITs will be checked by hand, and work may be rejected if there are too many errors.

**Question 1.** Briefly list all the objects visible in the image, separated by commas. If there are no objects in the image, write "none".

**Question 2a/3a.** How many [OBJECT]s are in the image?

**Question 2b/3b.** What color of [OBJECT]s are in the image? Choose all that apply.

**Question 2c/3c.** How realistic are the [OBJECT]s in the image? (1–3) Ignore the color of the object, and focus on any visible defects in the shape or structure.

**Question 4a.** Is the [OBJECT B] to the left or right of the [OBJECT A]? Provide your answer as a viewer of the image, not from the "camera's perspective".

**Question 4b.** Is the [OBJECT B] to above or below the [OBJECT A]? Provide your answer as a viewer of the image, not from the "camera's perspective".

Figure 11: Mechanical Turk annotation example.

