# OpenReview forum: "GenEval: An object-focused framework for evaluating text-to-image alignment"
_NeurIPS.cc/2023/Track/Datasets_and_Benchmarks — NeurIPS 2023 Datasets and Benchmarks Poster_

### Official Review · Reviewer_Pjnj · 2023-07-20
**A framework to evaluate generated images for the prompts they are created with**

**Rating:** 6
**Confidence:** 4

**Strengths:**

- A novel framework that is more accurate than CLIP based evaluation
- GenEval framework breakdowns the prompt into smaller elements to compute the final score on the generated content, whereas CLIP based approach computes a scalar indicating the image-text alignment
- Method is verified by comparing the results to those of annotators which labelled the generated images

**Additional Feedback:**

Authors unfortunately disclosed their identity and I wanted to flag this to AC: github link to the code has access to the primary author and acknowledgement includes identities that may have worked in the submission.

**Clarity:**

- It is a straight forward paper and easy to follow

**Correctness:**

- Evaluation method seems sufficient

**Documentation:**

- Code link is shared
- Evaluation dataset should be shared with their annotations for further comparisons

**Ethics:**

- There is no issue regarding ethical concerns

**Limitations:**

- For constant evaluation of the framework, labelling is required each time to measure the degradation of the GenEval tool. This could happen for a factor of reasons such as: bad generated content, insufficient object detector, etc.

**Opportunities For Improvement:**

- As authors said, object detectors may not be capable of detecting objects on generated images, and therefore finetuning might be necessary
- Quality of generated images have not been discussed. What if the quality of the generated content is not sufficient to make measurements

**Relation To Prior Work:**

- GenEval is more promising than CLIP based measurements and therefore may spur more research in this direction

**Summary And Contributions:**

Authors provide a framework to measure performance of image generators considering several aspects of the give prompts such as number of objects, object colors and their relative positioning. They use human labellers to compute the performance of the new framework and provide a comprehensive study of the framework.

---

> ### Author Response · Authors · 2023-08-23
> **Response to Reviewer Pjnj**
>
> Thank you for your thoughtful feedback. We address your specific concerns and questions below. Regarding **anonymity**: the call for papers for the Datasets and Benchmarks Track (https://nips.cc/Conferences/2023/CallForDatasetsBenchmarks) states:
>
> > Reviewing is in principle single-blind, hence the paper should not be anonymized.
>
> Since review for this track is not meant to be double-blind, we do not believe any error was made in disclosing identities.
>
>
> **Evaluating GenEval**
>
> Thank you for bringing up the matter of evaluating GenEval to ensure it is reliable for analyzing T2I models. We would like to clarify that part of the motivation of our approach is that the evaluation framework *does not* require constant human evaluation — in this paper (particularly in Section 4 and Figure 3, detailing the human study results) we verify that the GenEval tool largely agrees with human judgment along the specific axes that we consider, i.e., detecting objects, counting, measuring position, and classifiying color. This means that we can expect GenEval to be reliable as is, without needing constant evaluation. This also suggests that finetuning is not required within the scope that we focus on.
>
> Human re-evaluation will only be required in certain future use cases: for example, if GenEval is expanded to new tasks, upgraded to a new object detector model, or applied to a new domain (e.g. paintings).
>
>
> **Generated image quality**
>
> Thank you for bringing up the topic of generated image quality. It is true that our framework does not measure or evaluate the quality of generated images. We anticipate that GenEval will be used to augment existing metrics which do measure quality, e.g., FID score. However, from manual inspection we see that when image quality is poor, the object detector usually does not detect the required objects, and the image is likely to be marked incorrect. This is consistent with how a human might evaluate an AI-generated image. Overall, however, if the T2I model is prone to generate only low-quality images, then it may not be meaningful to evaluate its compositional capabilities.
>
> **Code and data release**
>
> We are in the process of making public all the necessary code and data for reproduction and usage of GenEval. Our prompt dataset, human annotations, and evaluation code are now accessible on our [GitHub repo](https://github.com/djghosh13/geneval), and our image generation code, data analysis, and documentation will be added soon.

---

> > ### Comment · Reviewer_Pjnj · 2023-08-24
> >
> > Thanks authors for responding and bringing the single-blind option to my attention. I stand by my rating for the paper and still believe that it is good enough for acceptance. I recommend authors to include in what scenarios human re-labelling may be required in the final manuscript.

---

### Official Review · Reviewer_nUMc · 2023-07-21

**Rating:** 5
**Confidence:** 4
**Correctness:** Basically accurate.
**Clarity:** Good.

**Strengths:**

1. The proposed method is more in line with the intuition of the evaluation, and the effect is better.
2. The proposed evaluation indicators and methods are very practical.
3. The author systematically proposes evaluation strategies and tasks, which are very helpful for evaluating the rationality of text-generated images.

**Additional Feedback:**

How did the author evaluate the positional relationship of the generated images? There is no corresponding expression in the paper.

**Documentation:**

The provided link does not exist.

**Ethics:**

Yes.

**Limitations:**

Not applicable.

**Opportunities For Improvement:**

See other feedback.

**Relation To Prior Work:**

It is reasonable to have a corresponding discussion.

**Summary And Contributions:**

Aiming at the difficulty of human evaluation on text-to-image generation tasks, the authors propose a new automatic evaluation framework, GenEval, which is an object-centric framework that can be used to evaluate attributes of combined images, such as object co-occurrence, location, count, and color. Compared with other automatic evaluation methods, it is more suitable for human evaluation.

---

> ### Author Response · Authors · 2023-08-23
> **Response to Reviewer nUMc**
>
> Thank you for your supportive comments. Please let us know if you have further questions or concerns about our paper.
>
> Regarding your clarification question:
>
> > How did the author evaluate the positional relationship of the generated images? There is no corresponding expression in the paper.
>
> We provide a description of the evaluation algorithms, including relative position, in Appendix C.3 of the supplementary material. To summarize, we compute the offset between two objects from their bounding box centroids, and determine whether this offset is greater than some minimum threshold so that the objects are perceived as not being fully overlapping/merged. Then the direction of this offset dictates whether the spatial relation is one of "above", "below", "left of", or "right of". Note that we are evaluating the relative position in 2D image space, and not the 3D world space of the scene.
>
> **Code and data release**
>
> We are in the process of making public all the necessary code and data for reproduction and usage of GenEval. Our prompt dataset, human annotations, and evaluation code are now accessible on our [GitHub repo](https://github.com/djghosh13/geneval), and our image generation code, data analysis, and documentation will be added soon.

---

### Official Review · Reviewer_Ygiv · 2023-07-21

**Rating:** 6
**Confidence:** 3
**Correctness:** The experimental settings are proper …
**Clarity:** This paper is well-written.

**Strengths:**

GENEVAL provides fine-grained reporting and analysis by evaluating the performance of text-to-image generation models on compositional tasks at the object level. This allows for a detailed analysis of failure patterns and can help identify areas for improvement.


**Additional Feedback:**

N/A

**Documentation:**

While the experimental settings are proper and the results are well presented, one limitation of the paper is that the dataset repo is not publicly available, making it difficult for others to reproduce the experiments and validate the results.

**Ethics:**

I do not see ethics issue in this paper.

**Limitations:**

Another drawback that exists is that the image obtained by the generative model may have some data distribution difference from the training data for target detection. And this is actually not considered in the article, and the current model of target detection may cause performance degradation due to the difference in data distribution.

**Opportunities For Improvement:**

1. Upgrade the GENEVAL framework when new compatible vision models emerge to improve its performance.
2. Try to consider if there will be a distribution gap between generated image and the training data for objection detection and if the gap will affect the performance of object detection models.

**Relation To Prior Work:**

Yes, this paper discusses the prior works.

**Summary And Contributions:**

The paper proposes GENEVAL, an object-centered framework for evaluating text-to-image alignment. It uses existing object detection and discriminative vision models to evaluate text-to-image generation model capabilities on compositional tasks.

---

> ### Author Response · Authors · 2023-08-23
> **Response to Reviewer Ygiv**
>
> We thank the reviewer for supporting our paper, confirming that our results are well-presented and our experimental settings are proper. We address your primary concerns below.
>
>
> **Data distribution difference**
>
> Thank you for bringing up the possibility of performance degradation in the object detector. For the current version of our evaluation framework, we find that the object detector we use is sufficient on the models we test, as shown by our human study results. To justify this claim in more detail, we compute agreement results for *object presence* (whether the specified objects are present in the image, ignoring further specifications like count, color, or position). We present our results in the table below, with a breakdown by image source.
>
> Note that CLIP retrieval (from LAION-5B) mostly returns real photo images that resemble the training distribution of the object detector. In this setting, we see a 5 percentage point drop in human agreement from interannotator to GenEval. By contrast, the T2I models SDv2.1 and IF-XL cause a 3 and 6 percentage point drop in agreement, respectively. These images are out-of-distribution for the object detector, but do not show significantly different results than for real-world images (CLIP retrieval). Thus, on the T2I models we tested, GenEval with our object detector model results in performance comparable to human crowdworkers, suggesting there is no serious performance degradation.
>
> | T2I Model      | Interannotator | GenEval |
> |:-------------- |:--------------:|:-------:|
> | CLIP retrieval |      0.90      |  0.85   |
> | SDv2.1         |      0.89      |  0.86   |
> | IF-XL          |      0.95      |  0.89   |
>
>
> However, finding the scenarios in which the object detector's performance degrades is indeed an interesting point of study, which we believe can be explored in-depth in future work. In particular, it may be useful to verify how well the object detector performs in a different domain of image generation, such as AI art/paintings.
>
>
> **Code and data release**
>
> We are in the process of making public all the necessary code and data for reproduction and usage of GenEval. Our prompt dataset, human annotations, and evaluation code are now accessible on our [GitHub repo](https://github.com/djghosh13/geneval), and our image generation code, data analysis, and documentation will be added soon.

---

### Official Review · Reviewer_ALyh · 2023-07-22
**New benchmark to evaluate text-to-image generative models on a set of predefined prompts using object detectors. Good benchmark but leaves some open questions.**

**Rating:** 7
**Confidence:** 4

**Strengths:**

- Method will improve when improved off-the-shelf object detectors and image-text similarity models are released.
- Interpretability, since the method provides an explanation of the detected failures.
- Solid coverage of related work
- The human study shows the method is more suited than existing methods.
- Good analysis of existing T2I models using the benchmark.

In general a helpful benchmark to test certain qualities of T2I models in detail.

**Additional Feedback:**

I will happily increasing my rating if the questions are clarified and especially the point in "Correctness" is addressed.

**Clarity:**

In general yes.

I have questions about the human study: How are the prompts selected and how many prompts are used? Why is the questions about the level of realism of the image asked, i.e. how is this annotation used later? How exactly are the numbers 77%, 83%, 88% calculated (formulae might help). The numbers described in line 215 (human ratings are unanimous) should have a corresponding figure or table with details.

Line 224: Why are exactly 533 prompts evaluated, where does this number come from? Since it is an automated method, why not evaluate more prompts?

**Correctness:**

Line 566 mentions about the hyperparameter c for relative position evaluation: "c = 0.1 optimizes human agreement". On which human agreement was this hyperparameter optimized? This is important: If you first collect the human agreement data and then optimize your hyperparameter to fit this human agreement, the agreement data is no longer an untouched test set and you might just overfit the human agreement data. Same question about line 550 (changing the object detector threshold from 0.3 to 0.9 to fit human agreement).

The baseline question is: Does your method actually agree well with humans in general, or does it just agree on this set of annotations because you optimized it on those?

**Documentation:**

Currently some details about the generated prompts are missing (see questions above) but in general this method is reproducible.

**Limitations:**

This method can only test the performance of the models on the 6 predefined prompt templates in table 1. This is a clear limitation compared to CLIPScore which can test performance on every possible prompt (however it is limited to the "correctness" of the returned score). I think this discussion is yet missing in the paper: How well do these 6 prompts cover the set of all possible prompts that users are likely to input? How could the template prompts be extended without making the method fail? "In the wild", prompts that users write often look like this "{{Prompt}}, wildlife photography, photograph, high quality, wildlife, f 1.8, soft focus, 8k, national geographic, award - winning photograph by nick nichols" (source: https://github.com/Dalabad/stable-diffusion-prompt-templates)

In line 32 the authors claim "current automated evaluation methods cannot analyze compositional capabilities" but in the related work chapter list "Dall-Eval [7] trains a task-specific object detector for
measuring each of several compositional reasoning tasks." (line 89) and "VISOR" which performs "evaluation over all triplets of object pairs and spatial relations" (line 97) which also seems like a compositional task. The statement on line 32 seems to contradict the discussed two related works.

**Opportunities For Improvement:**

The chosen baseline that the method compares to is CLIPScore. The CLIPScore used in this work is using CLIP-ViT-B-32 (line 582) which is a rather old (2021) and small model. What is the reason for using this baseline? Why not use any of the following:
* Stronger CLIP e.g. EVA-CLIP https://arxiv.org/abs/2303.15389 or EVA-CLIP-02 https://arxiv.org/abs/2303.11331 or any of the other models given by e.g. the open_clip package https://github.com/mlfoundations/open_clip or Vit-H-14 mentioned in line 252.
* Compare to the mentioned "VISOR" method on the spatial reasoning prompt

The same applies to the used CLIP for classifying object colors. It would be good to ablate a few CLIP models instead of simply chosing the original one, since it is already two years old and there may be better ones now.

Table 2 compares the models only using the new GenEval score. It would be good to also see the CLIPScore for these models to see whether the ranking of the methods changes between the two different scores.

**Relation To Prior Work:**

Yes

**Summary And Contributions:**

The paper proposes a new method to evaluate generative text-to-image (T2I) models like stable diffusion to overcome weaknesses of existing methods. The authors generate certain prompts with templates and evaluate the generated images using a combination of object detectors and a CLIP classifier. They validate their method with human evaluation. Finally they analyze various T2I models using their method.

---

> ### Author Response · Authors · 2023-08-23
> **Response to Reviewer ALyh (1/3)**
>
> Thank you for your thoughtful comments and in-depth feedback. In response to your comments, we have conducted several additional experiments. Our responses are detailed below.
>
> **Evaluation baselines**
>
> Thank you for your suggestions of additional baselines to compare against our evaluation framework. Since the introduction of CLIPScore, newer CLIP models with better overall performance have been released, which can be readily substituted for the original ViT-B/32 from 2021. We re-run the CLIPScore baseline with three more CLIP models:
> * ViT-L/14 from [Radford et al, 2021](https://arxiv.org/abs/2103.00020) (75.3% on ImageNet-1k);
> * ViT-H/14 from [Cherti et al, 2022](https://arxiv.org/abs/2212.07143) (78.0% on ImageNet-1k); and
> * EVA-02-CLIP E/14+ (the largest variant) from [Fang et al, 2023](https://arxiv.org/abs/2303.11331) (82.0% on ImageNet-1k).
>
> For comparison, the original ViT-B/32 obtains 63.2% accuracy on ImageNet-1k. The human agreement results are shown in the table below.
>
> We see that CLIPScore matches human judgment best when using the ViT-H/14, with particular improvement in the *two object* and *counting* tasks. Interestingly, however, we actually see a drop in performance in the position task compared to ViT-B/32. In addition, the EVA-02-CLIP model is significantly worse at the *two object*, *position*, and *color attribution* tasks --- all of which involve multiple types of objects. Overall, GenEval surpasses CLIPScore human agreement scores with all of the CLIP models, particularly on the complex compositional tasks.
>
> Reflecting our findings in this experiment, we will revise our paper to also include results using the strongest baseline, the OpenCLIP ViT-H/14.
>
>
> |     Metric     |  Overall  | Single object | Two object | Counting  |  Colors   | Position  | Color attribution |
> |:--------------:|:---------:|:-------------:|:----------:|:---------:|:---------:|:---------:|:-----------------:|
> | Interannotator |   0.878   |     0.990     |   0.872    |   0.901   |   0.854   |   0.797   |       0.853       |
> |    GenEval     | **0.834** |     0.953     |   0.799    | **0.823** |   0.808   | **0.823** |     **0.799**     |
> |    ViT-B/32    |   0.773   |   **0.991**   |   0.720    |   0.604   |   0.811   |   0.790   |       0.723       |
> |    ViT-L/14    |   0.760   |   **0.985**   |   0.692    |   0.583   |   0.796   |   0.786   |       0.719       |
> |    ViT-H/14    |   0.798   |   **0.986**   | **0.828**  |   0.734   | **0.827** |   0.682   |       0.732       |
> |  EVA-02-E/14+  |   0.600   |   **0.992**   |   0.628    |   0.612   |   0.804   |   0.240   |       0.322       |
>
>
> **Text-to-image model rankings**
>
> Thank you for suggesting a comparison of T2I models using CLIPScore to see if these results match the rankings by GenEval. The table below reports the average CLIPScore across images side-by-side with GenEval scores and overall human scores for the models that we had human annotations for. We will add this table to our paper.
>
> Overall, we find a consistent ranking of minDALL-E < CLIP retrieval < SDv1.5 < SDv2.1 < IF-XL. We do note, however, that the difference in CLIPScore between some pairs of models, notably SDv2.1 and IF-XL, is small in comparison to the relative performance difference according to human annotators or GenEval.
>
>
> | Model          | Human | GenEval | CLIPScore ViT-B/32 | CLIPScore ViT-H/14 |
> |:-------------- |:-----:|:-------:|:------------------:|:------------------:|
> | minDALL-E      |  ---  |  0.23   |        27.6        |        27.3        |
> | CLIP retrieval | 0.42  |  0.35   |        31.1        |        27.8        |
> | SDv1.5         |  ---  |  0.43   |        31.4        |        33.5        |
> | SDv2.1         | 0.57  |  0.50   |        32.1        |        36.2        |
> | IF-XL          | 0.72  |  0.61   |        32.4        |        36.5        |

---

> > ### Author Response · Authors · 2023-08-23
> > **Response to Reviewer ALyh (2/3)**
> >
> > **Hyperparameter tuning**
> >
> > Thank you for bringing up the important issue of hyperparameter tuning! While most of the design choices for our framework were made prior to the collection of crowdsourced human annotations, we did tune two hyperparameters --- the confidence threshold for the *counting* task and the distance threshold for the *position* task --- using the full set of human annotation data. Note that for fair comparison, we also tuned the threshold for CLIPScore on each task using the same data.
> >
> > To resolve this issue, we perform K-fold cross-validation on our existing annotations, as additional human annotation data is expensive to collect. We choose $K=5$ folds (each fold having a training/tuning size of 4800, and a validation size of 1200) to see the effects of tuning these two hyperparameters (and the CLIPScore thresholds) on smaller subsets of data. The validation set human agreement results are shown in the table below.
> >
> > Using K-fold cross-validation, the human agreement scores for GenEval do not drop by more than 0.001. Scores for CLIPScore do not drop by more than 0.01, suggesting that the optimal thresholds are fairly consistent for both methods. Further inspection shows that on the counting task, the chosen threshold is $C=0.9$ for all folds, while the chosen threshold on the position task is either 0.1 or 0.15 for all folds.
> >
> >
> > |                    | Counting task agreement | Position task agreement |
> > |:------------------ |:-----------------------:|:-----------------------:|
> > | GenEval            |    $0.823 \pm 0.013$    |    $0.822 \pm 0.013$    |
> > | CLIPScore ViT-B/32 |    $0.595 \pm 0.013$    |    $0.790 \pm 0.013$    |
> > | CLIPScore ViT-H/14 |    $0.734 \pm 0.013$    |    $0.680 \pm 0.018$    |
> >
> >
> > **Prompt templates**
> >
> > Thank you for bringing up the topic of our prompt templates, and how much coverage they have. We do not expect that the 6 tasks in the paper cover the diverse set of prompts that users may input. However, we do believe that they touch on several important image-text capabilities: understanding of exact terms (e.g. *counting*), relationships between objects (e.g. *position*), and binding between attributes/adjectives and objects (e.g. *color attribution*). Thus, our framework aims to provide a more targeted evaluation of specific aspects which are likely to be present in many users’ prompts.
> >
> > However, as in the example prompt you shared, many "in the wild" prompts consist of a description of the subject followed by a long list of modifiers. Future work can extend our prompt templates to cover more diverse prompts that resemble real user prompts. In the example you gave:
> > > "{{Prompt}}, wildlife photography, photograph, high quality, wildlife, f 1.8, soft focus, 8k, national geographic, award - winning photograph by nick nichols"
> >
> > a template from GenEval (e.g., "a photo of a blue cow") could be substituted into the "{{Prompt}}" to produce a more realistic test prompt, while leaving the metadata (i.e. what objects to expect) unchanged. Thus, the set of prompts for GenEval can be extended while leaving the rest of the framework untouched, perhaps by making use of prompt databases such as [here](https://huggingface.co/datasets/Gustavosta/Stable-Diffusion-Prompts). We hope to see future work expand on these possibilities to evaluate T2I models in more diverse settings.
> >
> > **Related work**
> >
> > Thank you for your positive comments about our coverage of related work. You are right that we had a contradiction in our wording with respect to alternative evaluation methods. Our intent was to state that *traditional* automated evaluation methods do not analyze compositional capabilities — most T2I models are often only evaluated with Inception score and FID score. We do recognize that the recent works of Dall-Eval [7] and VISOR [11] have both independently taken steps towards automated compositional evaluation along a similar vein as GenEval. We will fix this wording in our paper to clarify this point.

---

> > > ### Author Response · Authors · 2023-08-23
> > > **Response to Reviewer ALyh (3/3)**
> > >
> > > **Number of prompts**
> > >
> > > > Why are exactly 533 prompts evaluated, where does this number come from? Since it is an automated method, why not evaluate more prompts?
> > >
> > > Thank you for this question. In fact, the correct number is 553; we will fix this typo in the paper. For prompt generation, we randomly generated 100 prompts in each of the 6 tasks, sampling uniformly from the choices of objects, colors, counts, and positions. Afterwards, we filtered out 47 exact duplicate prompts, which left us with 553 prompts. We find that scores are consistent enough at this scale to conclusively compare the models tested, and scaling up the number of prompts requires a significant amount of compute, particularly for image generation.
> > >
> > >
> > > **Human study**
> > >
> > > > How are the prompts selected and how many prompts are used?
> > >
> > > For the human study, we selected 100 prompts at random from our full set of 553 prompts. For the three models we evaluated (SD v2.1, IF-XL, and CLIP retrieval) we had the crowdworkers annotate all 4 generated images for each of the 100 prompts, with 5 annotators per image. This resulted in a total of 6000 annotations on 1200 images.
> > >
> > > > Why is the questions about the level of realism of the image asked, i.e. how is this annotation used later?
> > >
> > > We asked annotators to mark the level of realism to investigate a potential relationship between object realism and the object detector’s confidence in classifying the object. Unfortunately, we did not find any strong correlation between the two.
> > >
> > > > How exactly are the numbers 77%, 83%, 88% calculated (formulae might help).
> > >
> > > The human agreement results consider whether each generated image is considered correct or incorrect, according to either the human annotators, GenEval, or thresholded CLIPScore. For each image, we measure *interannotator* agreement as the fraction of the other 4 annotators that the first annotator (randomly chosen) agrees with. We measure *GenEval and CLIPScore* agreement as the fraction out of all 5 annotators that the evaluation framework agrees with. This agreement is then averaged over all 1200 images to obtain the final agreement scores. We will revise our paper to include this more detailed description of computing human agreement.

---

> > > > ### Comment · Reviewer_ALyh · 2023-08-25
> > > >
> > > > Thank you for your insightful response. I believe these updates improve the paper and have decided to increase my rating to 7.

---

### Author Response · Authors · 2023-08-23
**Overall response**

We thank the reviewers for their time, constructive comments, thoughtful questions, and positive feedback. We are particularly grateful to the reviewers for highlighting the applications of our proposed evaluation framework for "interpretability" (ALyH), "detailed analysis of failure patterns" (Ygiv), and "evaluating the rationality of text-generated images" (nUMc). We also appreciate your positive comments on our paper being "easy to follow" (Pjnj) and our "solid coverage of related work" (ALyH).

One common question from multiple reviewers was regarding the **release of our code and data** to the public. At this point, we have now made the skeleton evaluation code and human annotation data publicly available on our [GitHub repo](https://github.com/djghosh13/geneval), and all the images that were annotated are also publicly accessible on [Huggingface](https://huggingface.co/datasets/djghosh/aigen-images). We are currently in the process of adding code and documentation for image generation and data analysis.

**We have addressed all specific reviewer feedback in detail in our comments below.** Several additional experiments and clarifications were suggested, which we will incorporate into our paper. We are grateful for all the feedback thus far, and are eager to further engage with the reviewers should more questions arise.

---

### Decision · Program_Chairs · 2023-09-22

**Decision:**

Accept (Poster)

**Comment:**

While the paper received mixed rating initially, the common theme of lack of public code and dataset, benchmarking for additional related baselines have been addressed by the authors and received acknowledgement from the reviewers. Based on this, I am recommending an Accept